# Parkin beyond Parkinson’s Disease—A Functional Meaning of Parkin Downregulation in TDP-43 Proteinopathies

**DOI:** 10.3390/cells10123389

**Published:** 2021-12-01

**Authors:** Katarzyna Gaweda-Walerych, Emilia Jadwiga Sitek, Ewa Narożańska, Emanuele Buratti

**Affiliations:** 1Laboratory of Neurogenetics, Department of Neurodegenerative Disorders, Mossakowski Medical Research Institute, Polish Academy of Sciences, 02-106 Warsaw, Poland; 2Neurology Department, St. Adalbert Hospital, Copernicus PL, 80-462 Gdansk, Poland; emilia.sitek@gumed.edu.pl (E.J.S.); gdansk.ewa1@gmail.com (E.N.); 3Division of Neurological and Psychiatric Nursing, Faculty of Health Sciences, Medical University of Gdansk, 80-211 Gdansk, Poland; 4Molecular Pathology Group, International Centre for Genetic Engineering and Biotechnology (ICGEB), AREA Science Park, 34149 Trieste, Italy; emanuele.buratti@icgeb.org

**Keywords:** TDP-43 pathology, parkin, mitophagy, fronto-temporal lobar degeneration (FTLD), amyotrophic lateral sclerosis (ALS), Parkinson’s disease (PD)

## Abstract

Parkin and PINK1 are key regulators of mitophagy, an autophagic pathway for selective elimination of dysfunctional mitochondria. To this date, parkin depletion has been associated with recessive early onset Parkinson’s disease (PD) caused by loss-of-function mutations in the *PARK2* gene, while, in sporadic PD, the activity and abundance of this protein can be compromised by stress-related modifications. Intriguingly, research in recent years has shown that parkin depletion is not limited to PD but is also observed in other neurodegenerative diseases—especially those characterized by TDP-43 proteinopathies, such as amyotrophic lateral sclerosis (ALS) and frontotemporal lobar degeneration (FTLD). Here, we discuss the evidence of parkin downregulation in these disease phenotypes, its emerging connections with TDP-43, and its possible functional implications.

## 1. Introduction

Frontotemporal dementia (FTD) is the second most common type of early onset dementia under 65 years of age, characterized by atrophy of the frontal and temporal lobes resulting in behavioral and/or language dysfunction [1,2,3]. Amyotrophic lateral sclerosis (ALS; alias motor neuron disease, MND) is an incurable neurodegenerative and neuromuscular disease, with degeneration of motor neurons in the brain and spinal cord which leads to progressive weakness of muscles, and gradual paralysis followed by respiratory failure [4,5,6]. A total of 10% of ALS cases and c.a. 30–40% of frontotemporal lobar degeneration (FTLD) cases are caused by genetic mutations, the remaining cases being sporadic [7]. Due to extensive clinical and genetic overlap, FTLD and ALS are thought to form a continuum of endophenotypes [8], and according to current diagnostic criteria, the term amyotrophic lateral sclerosis–frontotemporal spectrum disorder (ALS–FTSD) is used [9].

The common hallmark of sporadic and the majority of the genetic forms of ALS–FTSD is TDP-43 (transactive response DNA-binding protein 43 kDa) pathology, characterized by TDP-43 protein depletion from the nucleus and its cytoplasmic aggregation [10,11]. This pathology is present in the brain and spinal cord of 97% of ALS cases and c.a. 45% of FTLD cases [12,13].

Parkinson’s disease (PD) is a neurodegenerative disease with progressive death of dopaminergic neurons in the *substantia nigra*, the brain region producing neurotransmitter dopamine (DA). PD is clinically characterized by bradykinesia, resting tremor, and rigidity [14]. Around 15% of patients have genetic forms of the disease due to mutations in *PARK(1–23)* genes [15,16].

Mitochondrial dysfunction has been extensively described in sporadic and genetic forms of ALS–FTSD and PD [6]. In particular, the research on mitochondrial dysfunction in genetic PD caused by biallelic *PARK2* mutations [17] has gained pace since the discovery of parkin as a key mediator of mitophagy in 2008 [18] (Figure 1). Only two years earlier, in 2006, TDP-43 has emerged as an important protein for neurodegenerative diseases, such as ALS–FTSD, in which parkinsonian symptoms are reported [11,19] (Figure 1). Further, the observation of decreased parkin in TDP-43 proteinopathies is a relatively recent finding [20,21] (Figure 1), and its significance in terms of mitochondrial functioning has been hardly investigated. However, since these initial observations (Figure 1) a plethora of new research that has been performed is presented below.

In our review, we address the possible consequences of parkin deficit in TDP-43 proteinopathies. In the first part of the review, we provide background information on mitophagy, parkin/PINK1-associated PD, followed by a chapter on TDP-43 proteinopathy. As the main body of the article, we discuss the current evidence of parkin deficiency in sporadic and genetic forms of TDP-43 proteinopathies, its possible functional relevance, and the underlying molecular mechanisms. Further, we review the overlapping symptoms between PD and TDP-43 proteinopathies. Finally, we provide updated information on mitochondrial dysfunction in sporadic and major genetic forms TDP-43 proteinopathies and discuss the potential rationale for using parkin as a therapeutic target in ALS–FTSD.

## 2. Parkin and PINK1—The Key Players in Mitophagy Initiation

Mitophagy is a highly coordinated process whereby defective or old mitochondria are eliminated as whole organelles. This process occurs through the lysosomal pathway with the aid of the ubiquitin–proteasome system which concomitantly marks mitochondrial outer membrane proteins for disposal [22,23]. Mitophagy has been studied in numerous cellular and animal models either as a naturally occurring process or upon induction with various mitochondrial stressors [24]. Among mitophagy inducers, there are agents leading to mitochondrial membrane depolarization, mitochondrial respiratory complex inhibition, mutagenic stress, and proteotoxicity (e.g., CCCP, oligomycin, antimycin A, valinomycin, MPP+, and rotenone) [25,26].

The first step in mitophagy initiation is performed by mitochondrial kinase PINK1 (PTEN-induced kinase 1) “a sensor of mitochondrial damage” [26]. Under normal conditions, PINK1 expression levels are generally low in cells, because it is cleaved by protease PARL upon translocation to the inner mitochondrial membrane [27,28,29], followed by subsequent ubiquitination and degradation by the proteasome [30]. The other key player is represented by parkin (Parkin RBR E3 Ubiquitin Protein Ligase), which resides predominantly in the cytoplasm. Its expression levels in mitochondria are also normally low, as reviewed in Reference [31].

In response to mitochondrial stress, PINK1 normal processing is hindered. As a consequence, it gets anchored to the translocase of the outer mitochondrial membrane (TOMM) complex where subsequent PINK1 autophosphorylation leads to its auto-activation [32,33].

Further to this, PINK1 recruits parkin along with ubiquitin onto damaged mitochondria and activates them through phosphorylation on Ser65 [33]. To enhance this process, the Ser65-phosphorylated ubiquitin reinforces parkin activation and stabilization on the OMM [26].

Activated parkin ubiquitinates several mitochondrial outer membrane proteins (e.g., MFN2—mitofusin 2; TOMM20—translocase of outer mitochondrial membrane 20; and VDAC1—voltage dependent anion channel 1), tagging them for proteasome degradation with long ubiquitin chains [26,34]. Subsequent phosphorylation of these long ubiquitin chains by PINK1 on Ser65 makes it impossible for ubiquitin specific peptidase 30 (USP30) to detach ubiquitin residues from OMM proteins [35]. This generalized polyubiquitination of OMM proteins recruits, in turn, autophagy receptors, such as optineurin (OPTN) and sequestosome-1 (SQSTM1, p62), in such a way as to start forming an autophagosome around the damaged/old mitochondrion, eventually leading to its engulfment and subsequent digestion upon fusion with a lysosome [33].

## 3. Clinical Phenotypes Associated with Mutations in *PARK2* and *PARK6* Genes

### 3.1. Biallelic PARK2 (Parkin) and PARK6 (PINK1) Mutations Cause Young/Juvenile-Onset Parkinson’s Disease

Homozygous mutations in *PARK2* and *PARK6* are the most common known causes of autosomal recessive early onset parkinsonism (EOPD; age of onset—AAO < 50) and account for c.a. 5% of familial PD [17,36] (Figure 2A). EOPD has been further subdivided into cases with the onset before 21 years, grouped under the term of autosomal recessive juvenile parkinsonism (ARJPD, AAO < 21), and young-onset PD (YOPD, AAO > 21). Juvenile parkinsonism is very rare, patients have usually strong familial history, and present with atypical features. YOPD clinically resembles late-onset PD with positive family history in c.a. 3% cases [37]. EOPD cases have slower disease progression, less cognitive decline, and earlier motor fluctuations with dyskinesias and dystonia than sporadic PD (sPD) [38].

Patients with parkin mutations are good responders to anticholinergics and are very sensitive to small doses of levodopa that may cause severe dyskinesia and psychosis. Preserved olfactory function, sleep benefit, early postural instability, and brisk reflexes are often present [39]. Autonomic nervous system dysfunction and myocardial sympathetic denervation are less pronounced in *PARK2* mutation carriers than in individuals with sPD [40].

*PINK1* gene mutations cause YOPD with an onset between 30 and 50 years. Most patients have slow disease progression, good response to levodopa and levodopa-induced dyskinesias. Dystonia, hyperreflexia, sleep benefit, psychiatric problems, and dementia may be present early. Although most patients have features similar to those with sPD, a subset demonstrates features similar to those with biallelic *PARK2* mutations [41].

Clinical symptoms and postmortem pathology are both suggesting that the disease process is largely caused by the loss of dopaminergic nerve cells in the *Substantia nigra pars compacta* and subsequent nerve degeneration in the putamen with relative sparing of the *Locus coeruleus* and other brain regions. On the other hand, the neuropathology found in carriers of *PARK6* mutations is not well established, and it has also been noted that Lewy bodies’ pathology is inconsistently reported in cases with *PARK2* or *PARK6* mutations [42,43].

Finally, biallelic *PARK2* pathogenic variants have been very rarely associated with the phenotype of behavioral variant of FTD (bvFTD) with only mild parkinsonism [44].

### 3.2. Single Heterozygous PARK2 or PARK6 Mutations Lead to Subclinical PD

Single heterozygous *PARK2* or *PARK6* mutations occur with similar frequency in patients with sporadic PD and healthy individuals; their estimated penetrance is around 1–25%, and their role in pathogenesis is still not well understood [45,46,47,48] (Figure 2B). Individuals who carry a single *PARK2* or *PARK6* mutation are usually free of clinical motor symptoms; however, some of them can present clinical signs of parkinsonism [49,50] and may also have impaired facial emotion recognition [51]. Asymptomatic carriers of single *PARK2* or *PARK6* mutation show a stronger increase of cortical-motor-related activity during execution of self-initiated movements in functional magnetic resonance imaging (fMRI) studies, thus indicating additional recruitment of motor cortical areas during simple motor tasks, as is suggestive of compensatory response [52,53]. There is also evidence of nigrostriatal dysfunction in asymptomatic mutation carriers compared to control subjects: a significant reduction of striatal 18F-fluoro-L-DOPA uptake in putamen, caudate, dorsal, and ventral midbrain in positron emission tomography (PET) [50,54,55]. Subclinical deregulation of putamen dopaminergic signaling was observed in as much as 69% of asymptomatic single *PARK2* mutation carriers [50]. However, after a 5-year follow-up, these patients manifested no signs of parkinsonism at neurological examination and had very slow rates of nigrostriatal dysfunction progression, in comparison to patients with sporadic PD (a mean of 0.56% vs. 9–12% annual reduction in putamen F-DOPA uptake), suggesting that the probability of developing clinical parkinsonism by single *PARK2* mutation carriers is very low [56]. Finally, the reduced presynaptic dopamine terminal signaling is a pathological finding of subclinical nigrostriatal damage in the presence of one mutated parkin allele [55].

A possible molecular explanation for the observed abnormal nigrostriatal dysfunction in heterozygous carriers of *PARK2* or *PARK6* mutation is a reduction in enzymatic activity of ubiquitin ligase and kinase, respectively, leading to functional haploinsufficiency, unable to secure normal nigrostriatal activity [55,57]. Indeed, primary human skin fibroblasts and induced neurons derived from PINK1 p.Q456X heterozygotes demonstrate both reduced levels of PINK1 protein and kinase activity [57]. However, in a different scenario, a dominant-negative mechanism has been proposed for the PINK1 p.G411S mutation [57,58]. Heterozygous carriers of this mutation had normal levels of the PINK1 protein accompanied by a significant reduction in PINK1 kinase activity [57]. In addition, the authors observed impairment of ubiquitin phosphorylation by wild-type PINK1 in a heterodimeric complex (in vitro assay) [57].

Independently of the mechanism, those mutations likely interfere with the protective functions of the PINK1/parkin-mediated mitochondrial quality control [57]. In conclusion, therefore, even if recent findings support the notion of heterozygous *PARK2/PARK6* variants representing a strong risk factor for Parkinson’s disease [59], whether or not these heterozygotes with subclinical dysfunction will develop clinical parkinsonism over time is difficult to predict.

### 3.3. Parkin Inactivation in Sporadic PD

Parkin activity can be compromised by post-translational modifications, such as oxidative, nitrosative, or dopamine-related stress, which has implications for patients with sporadic PD and Lewy body neuropathology, as reviewed in References [31,60].

## 4. Loss-of-Function (LOF) and Gain-of-Function (GOF) Mechanisms in TDP-43 Proteinopathies

TAR DNA-Binding Protein-43 (TDP-43) was first identified in 2001 as a protein able to bind the HIV-1 TAR binding sequence [61], and in 2006, as the main component of aggregates found in the brains of patients with ALS and FTLD [11,19] (Figure 1). Recently, many reviews have been focused on elucidating the role of this protein in disease and normal development, and, for this reason, the reader is referred to these works for a more detailed description [10,62,63,64,65]. Briefly, TDP-43 belongs to the class of heterogeneous ribonucleoproteins (hnRNPs) that have been long referred to as the “RNA histones”. Normally, these proteins bind nascent RNA molecules and affect all aspects of RNA processing within the cell, from capping/splicing/polyadenylation to transport/translation and eventually degradation. At the structural level, TDP-43 possesses a highly structured N-terminus domain (NTD) [66,67] that controls protein dimerization/oligomerization [66,68]. This NTD is followed by two RNA Recognition Motifs (RRMs) that are responsible for sequence-specific binding to RNA [69,70] and then by an unstructured C-terminus region that plays a fundamental role in phase separation and aggregation [71,72,73,74].

In pathological aggregates, TDP-43 is subject to various post-translational modifications that include ubiquitination, phosphorylation, acetylation, sumoylation and is also cleaved to generate C-terminal fragments [62]. From a pathomechanistic point of view, there are two major disease pathways that have been proposed to become disrupted by TDP-43 aggregation and modifications: gain- and loss-of-function disease mechanisms [10]. The gain-of-function mechanisms may include various factors such as direct toxicity of the aggregates [75,76,77,78], direct toxicity of the C-terminal fragments [79,80], or indirect toxicity caused by sequestration of other proteins that are normally in close contact with TDP-43 in the cellular environment [81,82,83,84,85]. Intriguingly, it is also possible that aggregates might be protective at least during the early stages of the disease. This hypothesis is supported in a recent study based on random mutagenesis of the TDP-43 C-terminus where it has been observed that mutations that increase hydrophobicity and aggregation can decrease toxicity in yeast cells [85]. Alternatively, the loss-of-function pathological mechanisms might eventually occur through the sequestration of soluble TDP-43 in the aggregates, thus leaving not enough TDP-43 to perform its normal functions within cells. Therefore, this will result in multiple defects that range from preventing DNA damage to all aspects of RNA processing [86]. In support of this view, many recent studies agree that alterations in RNA metabolism could be a major contributor to ALS/FTLD processes in humans [87]. Most importantly, it should be kept in mind that all these gain- and loss-of-function possibilities do not necessarily exclude each other, and these different mechanisms could play a role at different stages of the disease. In conclusion, the emerging picture from all of these studies is that, following aggregation of TDP-43, a combination of RNA processing alterations might represent the principal disease contributor in patients with ALS and FTLD [88,89,90].

## 5. Interwoven Relations between TDP-43 and Parkin

Interestingly, apart from Parkinson’s disease, decreased levels of parkin have been found in several TDP-43 proteinopathies (Table 1/Figure 2C). To this date, in fact, there are several examples where the deregulation of parkin expression or its cellular localization has been linked to TDP-43 complex neuropathology and/or has been observed to occur following manipulations of TDP-43 expression levels [20,21,91,92,93,94] (Table 1).

### 5.1. Consistent Effects of TDP-43 Depletion on Parkin Levels

At the mechanistic level of RNA processing, TDP-43 has turned out to be crucial for the maintenance of brain enriched mRNAs with long introns (>100 kb) such as *PARK2* [20,21]. Indeed, *PARK2* pre-mRNA possesses multiple TDP-43 binding sites, suggesting that the control of its stability depends on TDP-43 RNA binding function, at least partially [20] (Figure 2C). In keeping with this view, and irrespectively of animal or cellular models under investigation, TDP-43 depletion consistently resulted in parkin mRNA/protein downregulation [20,21,91,94] (Table 1).

To date, a decrease in parkin mRNA has been observed upon TDP-43 depletion in mouse adult brain, stem cell-derived human neurons, HEK293T cells, human primary skin fibroblasts, and motor neurons obtained from patients with sporadic ALS [20,21,91,94] (Table 1). However, in motor neurons derived from patients with sporadic ALS, parkin decrease correlated with the presence of TDP-43 aggregates while ca. 95% of motor neurons without TDP-43 pathology showed normal parkin levels [21].

Recently, Sun et al., have elegantly shown that TDP-43 can affect parkin expression post-transcriptionally and in an intron/UTR-independent manner [92]. In a recent study, the authors have observed that overexpression of human TDP-43 in HEK293T cells downregulated plasmid-expressed, intron-free parkin, both at the mRNA and protein level (Table 1). This required both the RNA-binding and protein–protein interaction domains of TDP-43 that included the RNA recognition motif 1 (RRM1) and the glycine-rich domain (GRD) domain in the C-terminus [92].

In conclusion, the consistency of the results of TDP-43 depletion (Table 1) in different animal and cellular models and in patients with ALS/FTLD suggests that TDP-43 loss-of-function is crucial to maintaining parkin expression levels. Whether this is the only mechanism by which parkin decreases in ALS/FTLD remains elusive.

### 5.2. Discrepant Effects of TDP-43 Overexpression on Parkin Levels

Supporting the notion of parkin being a direct target for TDP-43, a significant increase in *PARK2* mRNA was observed in transgenic mice (hTDP43-Tg) brains compared to controls and a few cellular TDP-43 overexpression models [91,93,94]. Transgenic hemizygous mice harboring human TDP-43 A315T had increased levels of both soluble and insoluble parkin (by ca. 50 and 60%, respectively) [93] (Table 1). In contrast, other groups reported parkin downregulation upon wild-type and mutant TDP-43 overexpression, depending on the cellular model used [92,94,96] (Table 1).

Apart from parkin, TDP-43 ectopic expression influenced also other mitophagy key players. Sun et al. reported accumulation of cleaved PINK1(~52 kDa) insoluble aggregates in the cytoplasm via the mechanism of TDP-43 overexpression–related impairment of the proteasomal activity [92]. To confirm that these findings hold true also in vivo, the authors confirmed aggregation of cleaved endogenous PINK1 in the motor cortex of mice expressing Q331K mutant of TDP-43 [92]. Finally, they demonstrated that TDP-43-driven PINK1 accumulation affected negatively mitochondrial respiratory functions and lifespan in the *Drosophila* model [92].

At present, there is no satisfactory explanation for the observed differences between TDP-43 overexpression experiments. In contrast to TDP-43 knockdown, it should nonetheless be considered that TDP-43 overexpression levels may vary substantially between different labs/experimental systems. At the cellular level, this may result in the overexpressed TDP-43 binding/interacting with different partners, depending on the absolute level of overexpression reached during the study. If this could be the case, it would not be surprising that different overexpression levels could even have opposite effects on parkin expression. Of course, such a possibility would have to be experimentally tested and this is an area that certainly deserves further investigation.

### 5.3. Parkin as an E3-Ubiquitin Ligase Affects TDP-43 Aggregation

Notwithstanding the fact parkin levels are regulated by TDP-43, it has been shown that parkin, in turn, can affect the state of TDP-43 within cells. In particular, Hebron et al. have shown that parkin promoted Lys-48- and Lys-63-linked ubiquitination of TDP-43 and formed a multi-protein complex with histone deacetylase 6 (HDAC6) inducing sequestration of TDP-43 into cytosolic inclusions [93]. Co-expression of exogenous parkin and TDP-43 increased cytosolic co-localization of TDP-43, parkin, and ubiquitin. Moreover, parkin double knockout mice presented significantly higher levels of endogenous TDP-43 than control ones, although the number of hippocampal cells positive for TDP-43 was similar between transgenic and control mice [97].

### 5.4. Unanswered Question 1: Is Parkin Downregulation in TDP-43 Proteinopathies Functionally Relevant (Molecular Biology Perspective)?

While biallelic *PARK2* mutations lead to complete parkin depletion in patients with PD [98,99] (Figure 2A), the parkin decrease in carriers of single *PARK2* mutations remains yet to be determined (Figure 2B). In fact, it is important to note that in cellular PD models almost complete parkin deficiency (ca. 80%), obtained by silencing with small interfering RNAs (siRNA), is sufficient to trigger mitochondrial dysfunction in wild-type fibroblasts [100]. However, in contrast to patients with EOPD due to homozygous *PARK2* mutations, healthy carriers of heterozygous parkin mutation do not present abnormal mitochondrial function, including deregulated ultrastructure, morphology, and metabolism [99,101]. Likewise, 50% parkin silencing, which models haploinsufficiency (that would correspond to single *PARK2* mutation carriers), did not change any mitochondrial parameters, thus indicating the importance of the dose [100].

How do the above-mentioned observations correspond to TDP-43 proteinopathies? First of all, many transcriptomic/proteomic analyses in patients with ALS–FTSD or TDP-43 proteinopathy animal models do not report parkin expression alterations [102,103,104,105,106]. These might be due to the fact that all “omic” results (RNAseq, microarrays, and proteomics) carry a bias, since they are not able to distinguish between healthy cells and those with TDP-43 pathology. Moreover, the severity of TDP-43 pathology may be of varying grades [107,108].

In models where parkin expression changes are observed (Section 5.1 and Table 1), *PARK2* mRNA decrease upon TDP-43 depletion ranges from c.a. 25% in human neurons, 60% in human fibroblasts to 60–80% in rodent models. However, TDP-43 depletion does not mirror the complex pathology present in the brains of patients with ALS–FTLD. Thus, the local threshold effect might be extremely important, in terms of neurons with TDP-43 pathology and magnitude of parkin decrease. Furthermore, additional genetic or epigenetic risk factors may decide to what extent mitochondrial dysfunction would manifest itself.

Since carriers of single *PARK2* mutations can present subclinical brain dysfunction (see Section 3.2 for detailed description) it could therefore be speculated that subtle parkin decrease in TDP-43 proteinopathies can lead to similar subclinical phenotypes as in single *PARK2* mutation carriers. Nonetheless, this is an assumption that will need careful validation in the future. Indeed, there are limited existing data regarding parkinsonian phenotypes observed in TDP-43 proteinopathies, and this subject is discussed in Section 5.5 below.

### 5.5. Unanswered Question 2: What Is the Evidence of Parkinsonism in TDP-43 Proteinopathies (Clinical Perspective)?

ALS–FTSD comprises cases with dementia (ALS–FTD), cases with behavioral and/or cognitive impairment without dementia (ALSbci, ALSbi, and ALSci), ALS–parkinsonism/dementia complex (ALS–PDC, named also Western Pacific variant of ALS, lytico bodig), and other mixed variants [9].

ALS–FTSD is predominantly associated with TDP-43 proteinopathy [109], ALS being the most common one, and includes many movement disorders presentations, but MND and parkinsonian symptoms predominate.

First of all, it is important to keep in mind that cases of ALS–FTSD that have not been genetically defined may present other types of pathology (such as TAU, FUS, etc.). For this reason, in this review we focus on parkinsonism in genetic forms of ALS–FTSD with confirmed TDP-43 proteinopathy, caused by mutations in *C9orf72*, *PGRN*, and *TARDBP* [110,111,112,113].

Regarding FTLD it is interesting to note that the first links between this disease and parkinsonism extend before the recognition of TDP-43 proteinopathies. Indeed, until a few decades ago parkinsonian symptoms were regarded as a rare manifestation of FTLD associated with *MAPT* mutation, so the term frontotemporal dementia and parkinsonism linked to chromosome 17 (FTDP-17) was coined [114]. More recently, the association between progranulin mutations and parkinsonism [115,116,117] enabled us to distinguish between FTDP-17 (MAPT) and FTDP-17 (PGRN) [117]. Currently, the FTDP-17 term is no longer used clinically, considering that parkinsonism in FTLD has been shown to extend far beyond mutations in chromosome 17 [118].

Usually, FTLD is associated with either parkinsonian symptoms (e.g., in *PGRN* mutations) or with MND (e.g., in *C9orf72*).

In genetic FTLD parkinsonism is the most frequent movement disorder manifestation. It occurs in about 80% of cases. However, only ~ 1 in 3 cases with *C9orf72* mutation and 1 in 10 cases with *PGRN* mutations present with parkinsonism at onset [119]. According to other sources, depending on the population, parkinsonian symptoms occur in >20% of patients with ALS linked to *C9orf72* mutation, and as much as 50% to 75% of cases with bvFTD [120,121].

*PGRN* mutations are associated predominantly with parkinsonism and only occasionally with MND [122,123]. In some *PGRN* kindreds, parkinsonism has been reported in up to 80% of cases [124]. Importantly, patients with *PGRN* mutations demonstrate presynaptic dopaminergic deficit, as evidenced by Dopamine Transporter Scan (DAT-Scan) [125]. Unfortunately, DAT-Scan is not routinely performed, and parkinsonism in FTLD is usually described only based on clinical presentation.

It seems that *TARDBP* mutations may have a broader symptomatic spectrum than other gene mutations in FTLD, as they may be associated with all three conditions: ALS, FTLD, and parkinsonism [126].

At the clinical level, parkinsonism in FTLD syndromes is typically characterized by akinetic–rigid phenotype (symmetrical muscle rigidity, bradykinesia, hypokinesia, parkinsonian gait, and rarely resting tremor) [127] and may share clinical features with either progressive supranuclear palsy (PSP) or corticobasal syndrome (CBS) [128,129]. Sometimes multiple system atrophy-like presentations occur with dysautonomia, ataxia, and pyramidal symptoms [127].

Akinetic–rigid parkinsonism, as far as TDP-43 proteinopathies are concerned, occurs commonly in *PGRN* mutations and is rather uncommon in *C9orf72* mutations and *VCP* (Valosin Containing Protein) mutations and rare in *TARDP* mutations cases. PSP-like features may be rarely observed in both *C9orf72* and *PGRN* mutation carriers, CBS-like features are uncommon in *PGRN* and rare in *C9orf72* [127]. Of note, parkinsonian symptoms may be the first symptom in FTLD or develop after the occurrence of language or behavioral symptoms [121,130].

Among inherited FTLD cases, MND is observed mainly in patients carrying *C9orf72* or *TARDBP* gene mutations, but also those with *DCTN1* (Dynactin Subunit 1) and *VCP* gene mutations.

In some very rare clinical entities, such as Perry disease, caused by *DCTN1* mutation, TDP-43 pathology is predominant [131]. In others, it co-occurs with tau pathology (PSP, CBD) or alpha-synuclein pathology (PD, DLB) [13]. In Perry disease, parkinsonism, central hypoventilation, and weight loss [131] are accompanied by behavioral manifestations. The syndrome shares symptoms of PD, ALS and may fall into the FTLD spectrum [132].

Another example of the complex relationship between ALS and parkinsonism is the rare variant of ALS: ALS and parkinsonism/dementia complex (ALS/PDC) in which TDP-43 pathology may be accompanied by alpha-synuclein pathology. In ALS/PDC three types of pathology were described: the tauopathy-dominant type, the TDP-43 proteinopathy-dominant type, and the synucleinopathy-dominant type [133].

The frequency of parkinsonism in ALS–FTSD with TDP-43 proteinopathy cannot be easily established. Parkinsonism may be an under-diagnosed phenomenon in ALS–FTSD with TDP-43 proteinopathy due to several reasons. First, genetic testing and/or neuropathological examination is not routinely performed worldwide, remaining unproven in many cases. Secondly, *C9orf72* is a relatively recently described mutation as are Strong et al.’s criteria [9]. Third, at the diagnostic stage patients usually attend either a Dementia Clinic or Movement Disorders Clinics. Inevitably, some clinics focus mainly on cognitive/behavioral symptoms or motor symptoms and a mixed presentation of ALS–FTSD may be overlooked, especially if parkinsonism is not present at onset and develops later. At more advanced disease stages when the patients require constant supervision/care they are rarely seen by movement disorder specialists. Therefore, parkinsonism may not be diagnosed even in cases with pronounced akinetic–rigid manifestations. Furthermore, since akinetic–rigid parkinsonism is not a commonly known presentation of parkinsonism, specialists without movement disorder expertise are less likely to diagnose correctly someone that does not present with tremor phenotype.

In conclusion, the theme of parkinsonism in ALS–FTLD guarantees further research since at present there are a lot of gaps that need to be filled, specifically regarding parkinsonism in sporadic forms of ALS–FTSD with defined neuropathology and its clinical subgroups. Finally, it would be interesting to investigate whether parkinsonian symptoms observed in ALS–FTLD could be mediated by TDP-43-associated parkin decrease.

## 6. Additional Mechanisms of TDP-43—Mediated Mitochondrial Dysfunction

TDP-43 related mitochondrial dysfunction has been observed in genetic and sporadic forms of TDP-43 proteinopathies and in several cellular and animal models [134,135]. In particular, mitochondrial impairment ranged from abnormal mitochondrial ultrastructure, morphology, transport, respiration, membrane potential, dynamics, calcium buffering, and mitophagy, etc. (reviewed in References [136,137,138,139,140]).

Apart from the above-mentioned direct regulation of *PARK2* by TDP-43 (Section 5), potentially affecting mitochondria, numerous observations of enhanced mitochondrial localization of TDP-43 (either entire protein or its C- and N-terminal fragments) point to a global mechanism how TDP-43 can interfere with proper mitochondrial function and specifically mitophagy, as reviewed in Reference [138]. Consistently, overexpression models of TDP-43 and its C-terminal fragments display enhanced mitophagy, including parkin-mediated mitophagy [141,142]. Interestingly, the degradation of the C-terminal fragment of TDP-43 (CTF TDP-25) itself, localizing in mitochondria, is mitophagy-dependent [142].

Another aspect complicating the situation is that TDP-43 regulates other mediators of mitophagy and autophagy, as reviewed in References [140,143,144,145]. Specifically, TDP-43 imbalance (downregulation or upregulation) has been observed to impair autophagy flux and disrupt autophagosome–lysosomal fusion [106,140,145,146,147]. For example, overexpression of pathogenic CTF TDP-25 in mice leads to autophagy reduction/stalling [148]. Blocked autophagy, in turn, can result in the accumulation of depolarized mitochondria [149]. In this way, TDP-43-related autophagy deregulation can aggravate parkin/PINK1-dependent mitophagy dysfunction, as these processes are interconnected [150].

In addition, a majority of genes (*C9orf72*, *PGRN*, *TARDBP*, *OPTN* (optineurin), *TBK1* (TANK Binding Kinase 1), SQSTM1/p62, and *VCP* (Valosin Containing Protein)) that cause genetic forms of ALS–FTSD with TDP-43 proteinopathy encode proteins involved in mitophagy/autophagy [151].

As recent findings have demonstrated, enhanced TDP-43 mitochondrial localization could have broader consequences, through the activation of a robust immune response, which is a common characteristic observed in neurodegenerative diseases, including patients with PD or ALS–FTSD. At the mechanistic level, it has been shown that increased levels of TDP-43 in mitochondria cause the release of mtDNA into the cytoplasm, which switches on a sensor of cytoplasmic DNA, the cGAS/STING pathway, resulting in a neuroinflammatory response [152,153].

Importantly, the activation of the cGAS/STING pathway by circular cell-free mtDNA release, measured as an increase in proinflammatory Interleukin 6 (Il-6) levels has been confirmed in Parkinson’s disease patients with *PARK2/PARK6* mutations. A partial dose effect has been observed: symptomatic patients with biallelic mutations > symptomatic patients with single heterozygous mutation > asymptomatic subjects with single heterozygous mutations [59]. The lack of significant differences in ccf mtDNA/Il-6 levels between unaffected heterozygous mutation carriers and healthy control subjects or patients with sporadic Parkinson’s disease suggests that even slight parkin/PINK1 level/activity deregulation may be determinant in the transition from health to disease state.

### 6.1. How Different Genetic Backgrounds of TDP-43 Proteinopathies Might Modulate Mitophagy?

While it is well documented that TDP-43 pathology represents one of the main mechanisms underlying various aspects of mitochondrial dysfunction, its actions can also be modulated by different genetic backgrounds, such as mutations in *C9orf72*, *PGRN*, and *TARDBP* itself, that cause familial forms of ALS–FTSD.

### 6.2. Sporadic ALS/FTLD with TDP-43 Inclusions

In this section, we discuss sporadic ALS–FTSD cases where, apparently, TDP-43 proteinopathy is not modified by additional genetic factors.

First, it is important to note that induced pluripotent stem cells (iPSCs) and postmortem samples derived from patients with sporadic ALS and FTLD have demonstrated the destabilization of ribosomal and mitochondrial transcripts with decreased levels of corresponding mitochondrial proteins, as well as altered morphology and respiration [104,154]. A similar pattern of changes has been achieved by TDP-43 overexpression, suggesting a leading role for TDP-43 in the destabilization process [154].

Secondly, transcriptomic analysis of frontal cortex in 16 patients with sporadic FTLD–TDP followed by mRNA and protein level validation has demonstrated downregulation and decreased activity of several mitochondrial subunits of electron transport chain (ETC; complexes I, IV, and V) [104].

The results of these functional studies are supported by ultrastructural and morphological analyses which have revealed abnormal mitochondrial and lysosomal phenotypes in brain samples derived from patients with sporadic FTLD or ALS [155,156]. In some cases, mitochondria were observed to be aberrantly long with broken inner membrane and cristae, swollen matrix, and enhanced mitochondrial fusion with lysosomes [155,156]. In addition, decreased mitochondrial complex I activity, mitochondrial membrane potential, and mitochondrial ATP synthesis, along with elevated production of reactive oxygen species were also commonly observed in sporadic ALS/FTLD [156]. In this case, at the mechanistic level, it has been shown that TDP-43 activated the mitochondrial unfolded protein response (UPRmt) in both cellular and animal models [156].

Finally, it has been observed that TDP-43 aggregation-driven oxidative stress in mouse and human neurons depleted mitochondrial proteins encoded by nuclear-genome, leading to global mitochondrial dysfunction oxidative stress [157].

In conclusion, TDP-43-induced mitochondrial impairment certainly represents a critical aspect in TDP-43 proteinopathy.

### 6.3. C9ORF72 and TARDBP Mutations

In patients with *C9ORF72* hexanucleotide repeat expansion, TDP-43 pathology is accompanied by dipeptide repeat (DPR) protein aggregates arising from the unconventional repeat-associated non-ATG translation [158,159]. Similar to TDP-43 proteinopathy, *C9ORF72* pathology may act through loss-of-function (*C9orf72* haploinsufficiency) and gain-of-function mechanisms. The latter one is associated with the generation of long G4C2 expansions that may sequester cellular proteins from their normal functions and even generate toxic dipeptide repeat proteins [160].

From a pathological point of view, ALS- and FTLD-associated mutations in *C9ORF72* and *TARDBP* have shown similar effects on various mitochondrial parameters, such as increased oxidative and endoplasmic reticulum (ER) stress, impaired mitochondrial transport in axons, abnormal morphology, reduced ATP production, Ca2+ signaling, mitochondrial membrane potential, and activity of mitochondrial respiratory complexes [161,162,163,164,165,166,167], all reviewed in References [135,140]. Experimental models used to obtain these results comprised: autopsy brain samples, human primary skin fibroblasts, iPSCs-derived astrocytes, motor and cortical neurons of patients with *C9ORF72*- or *TARDBP*-linked ALS–FTSD, *Drosophila* and mouse models [161,162,163,164,165,166,167,168,169,170,171,172].

Abnormal mitochondrial phenotypes have been described in spinal motor neurons from patient-derived iPSCs with TDP43 S393L and G294V mutations [173], mice and *Drosophila* expressing TDP-43 A315T mutant [155,156].

In *Drosophila* muscle and patient fibroblasts with *C9ORF72* hexanucleotide repeat expansion, poly(GR) was observed to enter mitochondria and interfere with the function of the Mitochondrial Contact Site and Cristae Organizing System (MICOS), which led to abnormal mitochondrial inner membrane structure, metabolism, and ion homeostasis [171]. In iPSC-derived neurons, ectopic expression of DPR protein caused mitochondrial dysfunction through binding of mitochondrial ribosomal and respiratory complex V (ATP5A1) proteins [169,170]. This led to increased ATP5A1 degradation, accompanied by reduced ATP5A1 protein levels in cultured neurons and patients’ brains [169]. In addition, further research demonstrated that *C9orf72* is crucial for effective mitochondrial complex I assembly [167]. Finally, *C9orf72* hexanucleotide repeat expansion in astrocytes deregulated metabolism of adenosine, fructose, and glycogen, and the transport of mitochondria-specific energy substrates, thus increasing toxicity upon starvation [168,174].

In conclusion, it can therefore be speculated that the involvement of *C9ORF72* in mitochondrial complex assembly might be one of the reasons for observation of parkinsonisms in FTLD–*C9ORF72* cases, since mitochondrial complex dysfunction is one of the best-documented Parkinson’s disease molecular mechanisms [175].

### 6.4. PGRN Mutations Leading to Haploinsufficiency

As it has been well established in many previous studies, PGRN haploinsufficiency always leads to TDP-43 pathology [176,177]. Regarding mitochondria, one of the first transcriptomic analyses of PGRN-deficient primary human neurons showed downregulation of genes related to mitochondrial function, in particular, oxidative phosphorylation [178]. In addition to this evidence, mitochondrial membrane potential was higher in lymphoblasts derived from PGRN mutation carriers as compared to healthy controls upon 72 h serum starvation [179]. Furthermore, PGRN loss led to lipid metabolism deregulation, i.e., accumulation of polyunsaturated triacylglycerides, as well as a reduction of diacylglycerides and phosphatidylserines in fibroblast with *PGRN* mutations [180]. Considering that phosphatidylserine is a substrate for mitochondria-synthesized phosphatidylethanolamine, which is, in turn, important for membrane fusion, it is reasonable to hypothesize that this could affect the mitophagy process as a consequence.

Regarding the potential role played by PGRN in mitophagy, it has been recently reported that a decrease in PGRN exacerbated mitochondrial damage and dysfunction in podocytes from diabetic mice and that PGRN administration was able to restore mitophagy and mitochondrial biogenesis in podocytes challenged with high glucose through upregulation of Sirt1-PGC-1α/FoxO1 pathway [181].

Finally, as described in Section 5 and Table 1, we have recently demonstrated decreased parkin levels in fibroblasts carrying *PGRN* mutations which could not be rescued by TDP-43 overexpression, suggesting that established life-long TDP-43 pathology effects cannot be simply resolved through manipulation of TDP-43 level [94].

In summary, while the evidence on mitochondrial dysfunction in TDP-43 proteinopathies is substantial, it is not known whether TDP-43-related decrease of parkin could contribute to some of the observed mitochondrial phenotypes. To address this issue, it might be recommended that future studies on mitochondrial function in TDP-43 pathology model systems could test whether parkin is involved mechanistically, and, if yes, at which specific stage.

## 7. Increasing Mitophagy as a Therapeutic Approach for TDP-43 Proteinopathies

Mitophagy-enhancing therapies designed to increase PINK1 and parkin activity and/or inhibit ubiquitin-specific peptidase 30 (USP30) seem to be a plausible solution to rescue parkin/PINK1 deficiency in Parkinson’s disease caused by *PARK2* or *PARK6* mutations, as reviewed in Reference [35]. Notably, there is a report of natural compensation occurring in a biallelic *PARK2* mutation carrier that protected this individual from developing Parkinson’s disease even in his 80s [182]. The patient had upregulated levels of another mitophagy factor, NIP-3-Like Protein X (NIX), which bypassed the lack of functional parkin and restored functional mitophagy [182].

In TDP-43 proteinopathies, it has been shown that drugs increasing endogenous parkin levels were also able to decrease nuclear TDP-43 levels and prevent the neuronal loss and cognitive and motor decline in transgenic mice expressing either wild-type human TDP-43 or mutant A315T [97,183,184]. Likewise, upregulation of parkin and downregulation of PINK1 ameliorated the degenerative phenotypes of *Drosophila* expressing h-TDP-43 in neurons [92].

While parkin upregulation could ameliorate deleterious effects of TDP-43 proteinopathy in these cases, an opposite approach—overexpression of TDP-43—could not rescue decreased parkin levels in fibroblasts with *PGRN* mutations derived from FTLD patients, even if it increased PGRN levels [94]. Moreover, as already discussed, this is an issue that will require clarification in the future.

In contrast to precise targeting of the PINK1/parkin pathway, global mitophagy boost seems to aggravate phenotypes in TDP-43 proteinopathies. For example, in mice co-expressing TDP-43WT and mutant Q331K, rilmenidine administration induced autophagy and mitophagy, promoting accelerated nuclear TDP-43 clearance, but worsening the neurodegenerative phenotype, resulting in enhanced motor neuron death, and shortened lifespan [185].

Finally, in some genetic forms of ALS and FTLD with elevated basal levels of autophagy [151,180,186,187], mitophagy manipulations could have detrimental effects [35]. In conclusion, maintaining the delicate balance in mitophagy would be crucial for developing future therapies based on this mechanism.

## Figures and Tables

**Figure 1 cells-10-03389-f001:**
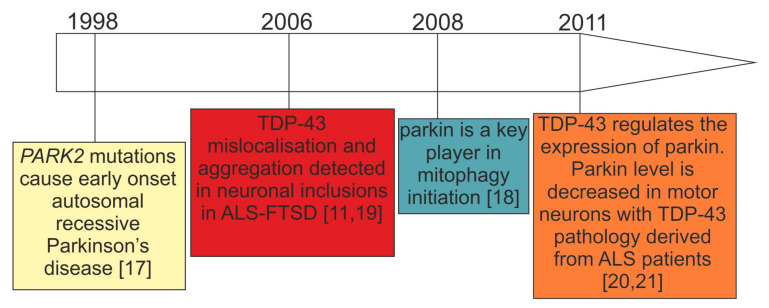
Timeline of the initial key research observations leading to the formation of our working hypothesis: parkin deficit observed in *PARK2*-related PD and patients with TDP-43 proteinopathies can lead to similar consequences.

**Figure 2 cells-10-03389-f002:**
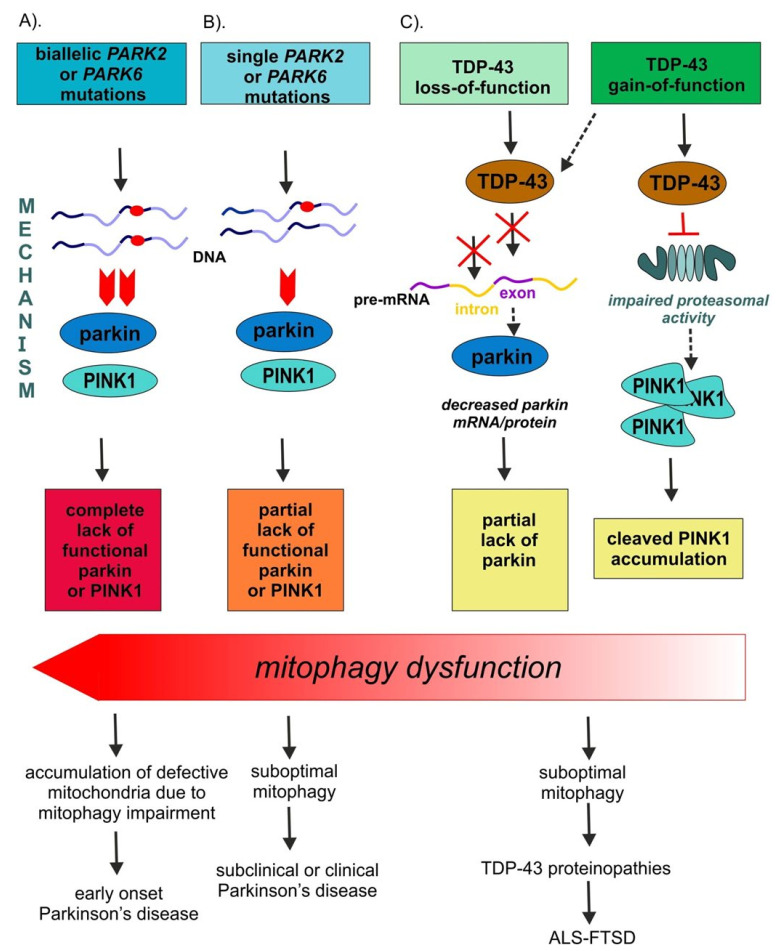
Schematic representation of parkin functional depletion grades. (**A**) Biallelic *PARK2* and *PARK6* mutations cause either complete lack of their respective proteins (parkin or PINK1) or their enzymatic activity, eventually causing early onset Parkinson’s disease. (**B**) Single heterozygous *PARK2* and *PARK6* mutations with partial lack of functional parkin or PINK1 are rarely found in symptomatic patients with PD and have mainly subclinical effects. (**C**) TDP-43 loss-of-function has been consistently observed to cause parkin downregulation, whilst the contribution of TDP-43 gain-of-function to parkin/PINK1 pathway deregulation requires more research (discussed in Section 5; see also Table 1).

**Table 1 cells-10-03389-t001:** Effects of TDP-43 pathology, TDP-43 depletion, or overexpression on parkin protein and mRNA levels.

TDP43-Proteinopathy Model	Cell Type/Treatment Length	Parkin mRNA/Protein	Accompanying Changes	References
Patients with sporadic ALS (n = 12) vs. control subjects	ca. 1000 motor neurons/-	Trend for decreased mRNA (microarray)	-	[95]
Patients with sporadic ALS (n = 11) vs. control subjects (n = 3)	Spinal cord motor neurons—only those with TDP-43 inclusions/-	Decreased protein (IF)	-	[21]
Carriers of *PGRN* mutations from families with FTLD	Human primary skin fibroblasts with *PGRN* mutations	Decreased mRNA/protein by ca. 60% (qRT-PCR)	Unchanged MFN2 and VDAC1 mRNA and protein	[94]
Mouse TDP-43 knockdown	Striatum injection of antisense oligonucleotides/2 weeks	Decreased mRNA by ca. 70% (RNAseq)	-	[20]
Mouse TDP-43 knockdown	Brain and spinal-cord injection of antisense oligonucleotides/2 weeks	Decreased mRNA by ca. 80% (qRT-PCR)	-	[21]
TDP-43 knockdown in human neurons (TDP-43 expression reduction by 60–75%)	Human neurons (iPSC-derived and HUES6 line) lentiviral shRNA constructs/na	Decreased mRNA by ca. 25% (qRT-PCR)	-	[21]
TDP-43 silencing (siRNA) in HEK293T	Human HEK293T (DMSO vs. mitochondrial uncoupler CCCP; siTDP-43 or si CTRL)/na	Decreased protein cytoplasmic localization (IF)	Decreased prohibitin 2 (PHB2)	[91]
TDP-43 silencing (siRNA) in skin fibroblasts derived from patients with FTLD	Human primary skin fibroblasts with *PGRN* mutations and control fibroblasts (siTDP-43 or siCTRL)/48 h	Decreased protein by ca. 40% (WB)	-	[94]
Overexpression of wild-type TDP-43-HA or mutant TDP-43-Q331K	Primary mouse neurons/motor cortex and human HEK293T cells/48 h	Decreased endogenous parkin mRNA and protein by c.a. 50% (qRT-PCR, WB)	Increased PINK1 protein	[92]
Exogenous co-expression of wild-type TDP-43-HA and intron-free human parkin or intron-free PINK1	Human HEK293T cells/48 h	Decreased intron-free parkin mRNA and protein by c.a. 50% (qRT-PCR)	Increased cleaved PINK1 protein forms insoluble cytoplasmic aggregates	[92]
Transgenic *Drosophila* knock-in of wild-type human TDP-43-H	Fly heads/na	Decreased mRNA and protein by c.a. 45% (qRT-PCR, WB)	-	[92]
Wild-type TDP-43 overexpression	Human HEK293T (DMSO vs. mitochondrial uncoupler CCCP; wild-type pLX-TDP-43-v5 vector/na)	Increased protein cytoplasmic localization (IF)	Increased prohibitin 2 (PHB2)	[91]
Wild-type TDP-43 overexpression	Human primary skin fibroblasts with transiently silenced PGRN (48 h) overexpressing wild-type flag-TDP-43 (24 h)	Increased protein by c.a. 40% (WB)	Increased PGRN protein	[94]
Wild-type TDP-43 overexpression	Human skin fibroblasts with *PGRN* mutations overexpressing wild-type flag-TDP-43 (48 h)	Decreased protein by c.a. 50% (WB)	-	[94]
Transgenic mouse with heterozygous knock-in of human mutant TDP-43 (A315T)	Whole-brain tissue	mRNA and protein reduced by 70% compared to wild-type controls (qRT-PCR, WB)	Abnormal neuronal mitochondrial cristae, fusion and fission defects;	[96]
Overexpressed wild-type TDP-43	Human M17 neuroblastoma cells	Increased protein by c.a. 50% (WB)	-	[93]
Transgenic mouse with knock-in of human mutant TDP-43A315T		Increased mRNA by c.a. 50% (qRT-PCR)	-	[93]

Abbreviations: IF—immunofluorescence, qRT-PCR—quantitative real-time PCR; iPSC—induced pluripotent stem cells, WB—Western blot. Experiments in Table 1 are presented as follows: dark grey-colored rows—evidence from patients with TDP-43 proteinopathies; light grey-colored rows: experiments with TDP-43 silencing; white smoke-colored rows—experiments with TDP-43 overexpression.

## Data Availability

No new data were created or analyzed in this study. Data sharing is not applicable to this article.

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
