# Peer review of "Parkin beyond Parkinson’s Disease—A Functional Meaning of Parkin Downregulation in TDP-43 Proteinopathies"

_cells, 2021, doi:10.3390/cells10123389_

Round 1

Reviewer 1 Report

Gaweda-Walerych, et al. have reviewed mechanistic links between parkin/PINK1- and TDP-43-related pathways. They suggested impairment of mitophagy and mitochondrial dysfunction to be a main outcome of the pathways. The discussion is comprehensive, powerful, and impressive. I have only single point before publication, as below;

1) Page 9, additional mechanisms of TDP-43-mediated mitochondrial dysfunction:

As the authors describe, autophagosome/endosome-lysosomal system is a main player  during mitophagy process. Recent studies have clarified pathomechanisms underlying autophagosome/endosome-lysosomal dysfunction in ALS/FTLD-TDP. I suppose that not only parkin/PINK1 depletion but also autophagosomal/endosomal dysfunction may be relevant to perturbation of mitophagy. If suitable, I recommend discussing about autophagosomal/endosomal dysfunction in the section ‘additional mechanisms of TDP-43-mediated mitochondrial dysfunction’. I feel that this make mitochondrial dysfunction in ALS/FTLD pathogeneses to be more clearly emphasized. In a yeast model, increased TDP-43 level disrupted the fusion and function of vesicles linked to autophagy-lysosome system (Leibiger, C, et al. Hum Mol Genet. 2018; 27: 1593-1607), suggesting that TDP-43 aggregation itself disturbs autophagosomal function. In addition, Hardy, et al. described that ALS/FTLD-related genes are often relevant to autophagy/endosome-lysosome function (ref. 85. Exp Neurol 2014; 262 Pt B: 75-83). For example, haploinsufficiency of C9orf72 correlated to disruption of autophagy, particularly of endosomal trafficking (Shi, Y, et al. Nat Med. 2018; 24: 313-325.). Impaired maturation of an endosome-lysosome pathway has been indicated autopsied patients with C9orf72 ggggcc hexanucleotide expansions (Riku Y, et al. Acta Neuropathol. 2019; 138: 783-793.).

Author Response

QUESTION#1

As the authors describe, autophagosome/endosome-lysosomal system is a main player during mitophagy process. Recent studies have clarified pathomechanisms underlying autophagosome/endosome-lysosomal dysfunction in ALS/FTLD-TDP. I suppose that not only parkin/PINK1 depletion but also autophagosomal/endosomal dysfunction may be relevant to perturbation of mitophagy. If suitable, I recommend discussing about autophagosomal/endosomal dysfunction in the section ‘additional mechanisms of TDP-43-mediated mitochondrial dysfunction’. I feel that this make mitochondrial dysfunction in ALS/FTLD pathogeneses to be more clearly emphasized. In a yeast model, increased TDP-43 level disrupted the fusion and function of vesicles linked to autophagy-lysosome system (Leibiger, C, et al. Hum Mol Genet. 2018; 27: 1593-1607), suggesting that TDP-43 aggregation itself disturbs autophagosomal function. In addition, Hardy, et al. described that ALS/FTLD-related genes are often relevant to autophagy/endosome-lysosome function (ref. 85. Exp Neurol 2014; 262 Pt B: 75-83). For example, haploinsufficiency of C9orf72 correlated to disruption of autophagy, particularly of endosomal trafficking (Shi, Y, et al. Nat Med. 2018; 24: 313-325.). Impaired maturation of an endosome-lysosome pathway has been indicated autopsied patients with C9orf72 ggggcc hexanucleotide expansions (Riku Y, et al. Acta Neuropathol. 2019; 138: 783-793.).

ANSWER#1

We thank the Reviewer for this comment. We realize/agree that autophagosome/endosome-lysosomal dysfunction may affect mitophagy initiation. Indeed, the accumulation of enlarged lysosomes has been observed in tissues of FTLD patients and we have cited appropriate papers (151,180,186,187) at the end of section 7 on the therapeutic approach. 

In this revised version, we have now added the suggested brief information in the section “additional mechanisms of TDP-43-mediated mitochondrial dysfunction”. In these additional comments, we have underlined that TDP-43 pathology regulates directly autophagy and that abnormal autophagy can affect mitophagy.

In parallel, we decided not to discuss autophagy dysfunction associated with genetic forms of FTSD. We are aware that mutations in C9orf72 or PGRN (to name to most common mutations in ALS-FTSD) cause autophagy dysfunction, and this could affect mitophagy as well. Nonetheless, we feel that the effects of C9orf72 or PGRN mutations are associated with many additional molecular effects beyond those caused by TDP-43 pathology itself, especially in the case of C9orf72 pathology. For this reason, the patients with these mutations might have many more processes deregulated (including mitophagy/autophagy) than those with sporadic ALS-FTSD.

Reviewer 2 Report

Gaweda-Walerych et al. reviewed mitochondrial and mitophagy dysfunctions in TDP-43 proteinopathy especially in relation to loss-of-function of parkin. This review well covered the outline and the latest evidences of that. It would be acceptable after the minor revision as shown below.

#1

“frontotemporal dementia (FTD)” is written twice in the introduction. In addition, the authors used two different terms; FTD and FTLD for the same disease entity. It would be better to explain the difference of them. At the first place, I recommend to use FTD only in the introduction to explain the epidemiology, and after that to use FTLD mainly as written in the reference; Ferrari et al., Neurobiol Aging 2019, 78, 98-110.

#2

The term “ALS-FTSD spectrum” seems weird, because there are two “spectrum” in it. “ALS-FTLD spectrum” or “ALS-FTSD” would be better.

#3

Line 14, page 1.

“Parkin along with PINK1 are key regulators” should be revised to “Parkin along with PINK1 is a key regulator” or “Parkin and PINK1 are key regulators”.

#4

Line 196-197, page 6.

Only abbreviations (ALS and FTLD) should be used. They are already written in the introduction.

#5

Line 325-327, page 9.

It should be revised to “between unaffected heterozygous mutation carriers and healthy control subjects or patients with idiopathic Parkinson’s disease”

#6

Line 334, page 9.

TARDPB” should be revised to TARDBP.

#7

Line 366, 383, 392, page 10. Line 468, page 12

“C9ORF72 expansion” may not be an appropriate term. C9ORF72 hexanucleotide repeat expansion or C9ORF72 mutation would be better.

#8

Line 368, page 10.

“Similarly to TDP-43 proteionpathy” would be better.

#9

Line 467, page 12.

“parkinsonism” or “parkinsonian symptoms” would be more natural than “PD symptoms” in this context.

#10

Line 492, page 12.

In general, MND is mainly sporadic. “In genetic FTLD parkinsonism” or some other appropriate phrases should be added before “MND”.

Author Response

QUESTION#1 “frontotemporal dementia (FTD)” is written twice in the introduction. In addition, the authors used two different terms; FTD and FTLD for the same disease entity. It would be better to explain the difference of them. At the first place, I recommend using FTD only in the introduction to explain the epidemiology, and after that to use FTLD mainly as written in the reference; Ferrari et al., Neurobiol Aging 2019, 78, 98-110.

ANSWER#1 We have explained both terms and in this revised version we have now used mainly the term  FTLD, in line with the Reviewer’s suggestion. In this version, we have then used FTD only when referring to a behavioral variant of FTD (bvFTD) in accordance with Rascovsky et al.’s criteria.

QUESTION #2 The term “ALS-FTSD spectrum” seems weird, because there are two “spectrum” in it. “ALS-FTLD spectrum” or “ALS-FTSD” would be better.

ANSWER#2 We have corrected this.

QUESTION #3 Line 14, page 1.“Parkin along with PINK1 are key regulators” should be revised to “Parkin along with PINK1 is a key regulator” or “Parkin and PINK1 are key regulators”.

ANSWER#3 We have corrected this in the abstract.

QUESTION #4 Line 196-197, page 6. Only abbreviations (ALS and FTLD) should be used. They are already written in the introduction.

ANSWER#4 We have corrected this throughout the whole text.

QUESTION #5 Line 325-327, page 9. It should be revised to “between unaffected heterozygous mutation carriers and healthy control subjects or patients with idiopathic Parkinson’s disease”

ANSWER#5 We have corrected this according to the Reviewer’s suggestion.

QUESTION #6 Line 334, page 9. “TARDPB” should be revised to TARDBP.

ANSWER#6 We have corrected this error according to the Reviewer’s suggestion.

QUESTION #7 Line 366, 383, 392, page 10. Line 468, page 12

“C9ORF72 expansion” may not be an appropriate term. C9ORF72 hexanucleotide repeat expansion or C9ORF72 mutation would be better.

ANSWER#7 We have corrected this according to the Reviewer’s suggestion.

QUESTION #8 Line 368, page 10. “Similarly to TDP-43 proteinopathy” would be better.

ANSWER#8 We have corrected this according to the Reviewer’s suggestion.

QUESTION #9 Line 467, page 12. “parkinsonism” or “parkinsonian symptoms” would be more natural than “PD symptoms” in this context.

ANSWER#9 We have used the term parkinsonian symptoms in line with the Reviewer’s suggestion.

QUESTION #10 Line 492, page 12. In general, MND is mainly sporadic. “In genetic FTLD parkinsonism” or some other appropriate phrases should be added before “MND”.

ANSWER#10 We have now corrected this: “Among inherited FTLD cases MND is observed mainly in patients carrying”

Reviewer 3 Report

In this review Gaweda-Walerych and colleagues promise a interesting take on the role of Parkin in TDP-43 proteinopathies. The review is well documented and researched, and the topic promised by the title offers a new view on TDP-43 associated neuropathologies.

Unfortunately, the title is slightly misleading and the review fall short on its promise. There is two major parts in the review with an up-to-date statement on the role of Parkin in PD and associated mitochondrial dysfunctions. A second part on TDP-43 proteinopathies and associated mitochondrial dysfunctions. However the link between these two part is not easy to grasp. The last part, which offers a dive in the unanswered questions in the field, is more in line with the title of this review.

In general the review is well written but there is no story line. The writing style is "fact (citation). next fact (citation).". The review would benefit from some reshuffling and reorganization to make it easier to follow and more appealing for the reader. It would also be better to really focus on the topic promised by the title instead of listing other unrelated facts.

As a suggestion, some of the questions asked in the last part could potentially be answered using available datasets from RNAseq or proteomics from relevant models for the review (either PD or TDP-43 pathology), to see if some of the question asked could actually be partially answered with these already existing datasets. This would bring a significant value to the review and would appeal more to both field of PD and TDP-43 proteinopathies.

In general, this work bring something new in the field of neurodegenerative diseases, but would deserve more work to really differentiate itself from other existing review on similar topic.

Minor comment, there is a point 2.1 but no 2.2...

Author Response

QUESTION #1

Unfortunately, the title is slightly misleading and the review fall short on its promise.

ANSWER#1: The Reviewer has raised some important points and we have now substantially modified the general architecture of the review in several ways. First, section 5 has been substantially revised now. It is now the longest section of the manuscript and it is entirely dedicated to the title “Parkin beyond Parkinson’s disease - a functional meaning of parkin downregulation in TDP-43 proteinopathies”. Second, we have added new information regarding parkin expression levels in Table 1, from RNAseq, microarrays, and qRT-PCR data. Finally, we have rearranged section 5, moving to this section “unanswered questions” from former section 7, as we feel it naturally summarizes the findings presented through sections 5.1-5.3. All the changes are highlighted in yellow.

QUESTION #2

There are two major parts in the review with an up-to-date statement on the role of Parkin in PD and associated mitochondrial dysfunctions. A second part on TDP-43 proteinopathies and associated mitochondrial dysfunctions. However the link between these two part is not easy to grasp.

ANSWER#2: We thank the Reviewer for pointing out the parts that might have not been clear enough for readers.

To make the aim of our review clearer for readers,  we have now extended/improved the explanation in the Introduction section adding a new Fig 1 which presents the timeline of key research findings that prompted us to form the main hypothesis of this review. Indeed, till 2011 year (Polymenidou et al 2011) it was not known that parkin is one of the TDP-43 targets and that TDP-43 depletion could lead to downregulation of parkin.

Moreover, it must be considered that researchers that work in the field of parkin might not be very familiar with TDP-43 biology, and vice versa. That is why we believe that short informative sections (3 and 4) are important to provide essential information for readers from different backgrounds.

Please note, that the review offers two perspectives while describing the relations between TDP-43 and parkin: molecular biology perspective (parkin: section 2; TDP-43:section 4) and clinical perspective (parkin-related PD: section 3, and parkinsonisms in ALS-FTSD: section 5.5).

At the molecular level, the decrease of parkin could be potentially one of the driving mechanisms responsible for mitochondrial dysfunction in TDP-43- proteinopathies. However, although many studies document mitochondrial dysfunction in TDP-43- proteinopathies (section 6), none of these studies attribute it to parkin deregulation. For this reason, section 6 is in our view important and cannot be omitted. Nonetheless, we have added a short summary at the end of section 6 to make this issue more clear to readers.

Finally, the Reviewer is right that the link between these two parts is not easy to grasp - we indeed wanted to convey that the understanding of the role of parkin in TDP-43 proteinopathies and any overlapping similarities with parkin-driven PD is still very weak. We hope that our review will inspire researchers to look more deeply into this problem, by designing appropriate experiments, and for this reason, we drew considerable attention to existing discrepancies in the published data (see e.g. section 5.2).

In section 7.2 “Evidence of parkinsonism in TDP-43 proteinopathies”) we highlight the fact that parkinsonisms in the ALS-FTLD spectrum are underestimated and barely described.

Thus, our conclusion is the link between parkin and TDP-43 proteinopathies is still not well established and remains a potentially important open research question.

QUESTION #3

The last part, which offers a dive in the unanswered questions in the field, is more in line with the title of this review.

ANSWER #3 We thank the Reviewer for this comment. We have now moved section “Unanswered questions: is parkin downregulation in TDP-43 proteinopathies functionally relevant (molecular biology perspective)?” as section 5.4, where it certainly fits better (see also ANSWER#1).  

QUESTION #4

In general the review is well written but there is no story line. The writing style is "fact (citation). next fact (citation).". The review would benefit from some reshuffling and reorganization to make it easier to follow and more appealing for the reader. It would also be better to really focus on the topic promised by the title instead of listing other unrelated facts.

ANSWER #4 According to the Reviewer’s suggestions, in the revised version we have now substantially changed the layout of the review (see also ANSWER #1 and ANSWER #2) and all the changes are highlighted in yellow. Referring to sections 6.1-6.4, while there is no clear evidence that any of these reported mitochondrial abnormalities could be attributed to parkin decrease, we felt it was important to provide an updated view on mitochondrial dysfunction observed in TDP-43 proteinopathies, including selected genetic forms of ALS-FTSD. 

QUESTION #5

As a suggestion, some of the questions asked in the last part could potentially be answered using available datasets from RNAseq or proteomics from relevant models for the review (either PD or TDP-43 pathology), to see if some of the question asked could actually be partially answered with these already existing datasets. This would bring a significant value to the review and would appeal more to both field of PD and TDP-43 proteinopathies.

ANSWER #5 We agree with that observation. We have now added new information regarding parkin expression levels in Table 1, from RNAseq, microarrays, and qRT-PCR. However, it is worth noting that all “omic” results (RNAseq, microarrays, proteomics) carry a certain bias since they are not able to distinguish between healthy cells and those with TDP-43 pathology. This can be crucial, as according to Lagier-Tourenne et al. (2012), parkin decrease in motor neurons derived from sporadic ALS patients correlates strongly with the presence of TDP-43 aggregates while c.a. 95% of motor neurons without TDP-43 pathology show normal parkin levels.

QUESTION #6

Minor comment, there is a point 2.1 but no 2.2...

ANSWER #6  The numbering has been corrected and double-checked throughout the revised manuscript.

Round 2

Reviewer 3 Report

The review significantly improved with the reshuffling of the information and the small information added.

Minor comments:

  • It would be nice to replace "ALS-FTLD patients" "PD patients" "ALS patients" etc... by Patient with ALS-FTLD/PD/ALS. This is more respectful for the patients, as they are not characterized by their pathology, but are patients with a pathology.
  • The timeline in Figure 1 does not seem complete... I know the authors wish to not spoil their manuscript for the reader at the beginning, but there is some missing information that could be added within this timeline (including the one found in table 1) to show that the research on the link between Parkin and TDP-43 did not stop in 2011 (ref 92 for example)... It would be a good summary tool for the reader to follow the story line.

Author Response

QUESTION #1

It would be nice to replace "ALS-FTLD patients" "PD patients" "ALS patients" etc... by Patient with ALS-FTLD/PD/ALS. This is more respectful for the patients, as they are not characterized by their pathology, but are patients with a pathology.

ANSWER #1

According to the Reviewer’s suggestions, we have now made appropriate corrections in the manuscript and they are highlighted in blue (to distinguish round 1 from round 2).

QUESTION #2

The timeline in Figure 1 does not seem complete... I know the authors wish to not spoil their manuscript for the reader at the beginning, but there is some missing information that could be added within this timeline (including the one found in table 1) to show that the research on the link between Parkin and TDP-43 did not stop in 2011 (ref 92 for example)... It would be a good summary tool for the reader to follow the story line.

ANSWER #2

We thank the Reviewer for this comment about Figure 1. We have now made it clear that Fig 1 refers only to the initial key research observations in the four mentioned fields. We have changed the caption of Fig. 1 to explain this and added a sentence in the Introduction. The changes are highlighted in blue. From our point of view, we cannot add only reference 92 to Figure 1 as we would have to refer to all the important research articles that had contributed to the subject of our review. All relevant research articles are summarized in Table 1 and discussed in section 5.
